

# The invasive land planarian *Platydemus manokwari* (Platyhelminthes, Geoplanidae): records from six new localities, including the first in the USA

Jean-Lou Justine[1], Leigh Winsor[2], Patrick Barrière[3], Crispus Fanai[4], Delphine Gey[5], Andrew Wee Kien Han[6], Giomara La Quay-Velázquez[7], Benjamin Paul Yi-Hann Lee[8,9], Jean-Marc Lefevre[10], Jean-Yves Meyer[11], David Philippart[12], David G. Robinson[13], Jessica Thévenot[14] and Francis Tsatsia[4]

[1] Institut de Systématique, Évolution, Biodiversité, ISYEB, UMR7205 CNRS, EPHE, MNHN, UPMC, Muséum National d'Histoire Naturelle, Sorbonne Universités, Paris, France
[2] College of Marine and Environmental Sciences, James Cook University, Townsville, Australia
[3] Conservatoire d'Espaces Naturels de Nouvelle-Calédonie (CEN NC), Pôle Espèces Envahissantes (PEE), Koné, New Caledonia
[4] Biosecurity Solomon Islands, Ministry of Agriculture and Livestock, Honiara, Solomon Islands
[5] Service de Systématique Moléculaire, Muséum National d'Histoire Naturelle, Paris, France
[6] SingHealth Polyclinics, Marine Parade, Marine Parade Central, Singapore
[7] Department of Biology, Universidad de Puerto Rico, San Juan, Puerto Rico, USA
[8] Durrell Institute of Conservation & Ecology, University of Kent, United Kingdom
[9] National Parks Board, Singapore
[10] Environnement et Cadre de Vie, Ville de Caen, Caen, France
[11] Délégation à la Recherche, Gouvernement de la Polynésie française, Papeete, Tahiti, French Polynesia
[12] FREDON de Basse Normandie, Hérouville-Saint-Clair, France
[13] USDA APHIS National Malacology Laboratory, Academy of Natural Sciences, Philadelphia, PA, USA
[14] Service du Patrimoine Naturel, Muséum National d'Histoire Naturelle, Paris, France

Corresponding author
Jean-Lou Justine, justine@mnhn.fr

## ABSTRACT

The land planarian *Platydemus manokwari* de Beauchamp, 1963 or "New Guinea flatworm" is a highly invasive species, mainly in the Pacific area, and recently in Europe (France). We report specimens from six additional countries and territories: New Caledonia (including mainland and two of the Loyalty Islands, Lifou and Maré), Wallis and Futuna Islands, Singapore, Solomon Islands, Puerto Rico, and Florida, USA. We analysed the COI gene (barcoding) in these specimens with two sets of primers and obtained 909 bp long sequences. In addition, specimens collected in Townsville (Australia) were also sequenced. Two haplotypes of the COI sequence, differing by 3.7%, were detected: the "World haplotype" found in France, New Caledonia, French Polynesia, Singapore, Florida and Puerto Rico; and the "Australian haplotype" found in Australia. The only locality with both haplotypes was in the Solomon Islands. The country of origin of *Platydemus manokwari* is New Guinea, and Australia and the Solomon Islands are the countries closest to New Guinea from which we had specimens. These results suggest that two haplotypes exist in the area of origin of the species, but that only one of the two haplotypes (the "World haplotype") has, through human agency, been widely dispersed. However, since *P. manokwari*

is now recorded from 22 countries in the world and we have genetic information from only 8 of these, with none from New Guinea, this analysis provides only partial knowledge of the genetic structure of the invasive species. Morphological analysis of specimens from both haplotypes has shown some differences in ratio of the genital structures but did not allow us to interpret the haplotypes as different species. The new reports from Florida and Puerto Rico are firsts for the USA, for the American continent, and the Caribbean. *P. manokwari* is a known threat for endemic terrestrial molluscs and its presence is a matter of concern. While most of the infected territories reported until now were islands, the newly reported presence of the species in mainland US in Florida should be considered a potential major threat to the whole US and even the Americas.

## INTRODUCTION

The land planarian *Platydemus manokwari* de Beauchamp, 1963 or "New Guinea flatworm" is an invasive species, recorded in 15 countries in the World, and recently in France in a hothouse (*Justine et al., 2014*). *Platydemus manokwari* is the only flatworm listed in the "100 world's worst invasive alien species" (*Lowe et al., 2000*); it is a predator of land snails and is considered a danger to endemic snails wherever it has been introduced. Its distribution records, reproduction, biology, prey lists, impacts, and possible control options were recently reviewed (*Justine et al., 2014*).

Alien land planarians, generally originating from the Southern Hemisphere or from tropical Asia, are now found in all parts of the world (*Álvarez-Presas et al., 2015*; *Álvarez-Presas et al., 2014*; *Breugelmans et al., 2012*; *Cannon et al., 1999*; *Jones, 1998*; *Jones, 2005*; *Justine, Thévenot & Winsor, 2014*; *Kawakatsu et al., 2002*; *Lago-Barcia et al., 2015*; *Mateos et al., 2013*) and some of these more invasive flatworms pose a threat to local species that are included in their prey, including earthworms (*Boag et al., 1994*; *Boag & Yeates, 2001*; *Jones et al., 2001*; *Murchie & Gordon, 2013*) and snails (*Winsor, Johns & Barker, 2004*).

We report here the presence of *P. manokwari* in several additional countries and territories: Singapore, New Caledonia (including mainland and two of the Loyalty Islands), an additional island in French Polynesia, Wallis and Futuna (from two of the islands, Uvea and Futuna), the Solomon Islands, Puerto Rico and Florida, USA—the latter being the first records on the American continent. We show that barcodes of specimens comprise two haplotypes, one found in many localities in the world and one found in Australia. Specimens from the Solomon Islands were the only ones to show genetic diversity, with both haplotypes present in the same locality.

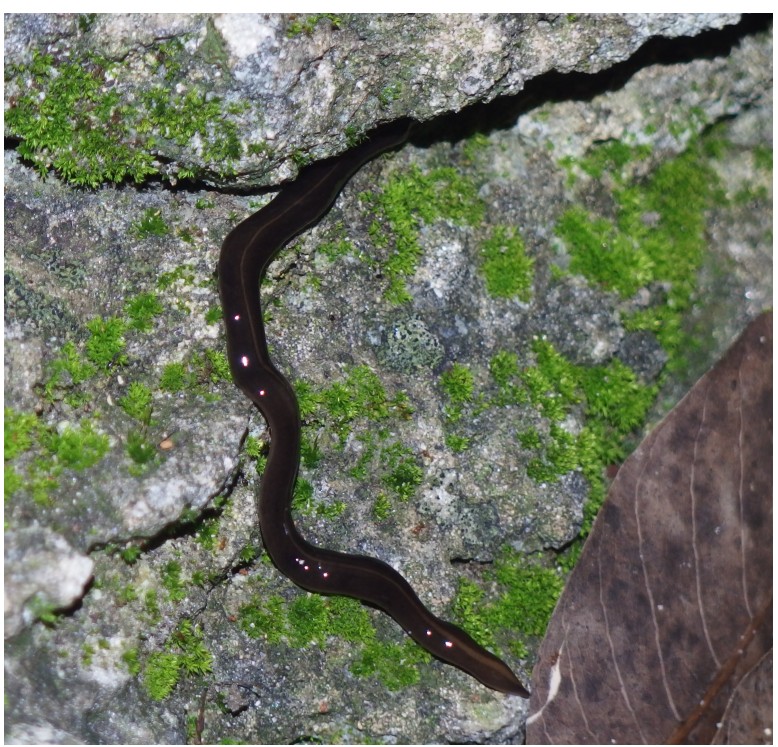

**Figure 1** *Platydemus manokwari* **in Coral Gables, Florida, USA.** Photograph by Makiri Sei.

## MATERIAL AND METHODS

### Origin of new reports

The new findings reported here were collected thanks to a worldwide campaign of several steps: (a) an article published in this journal (*Justine et al., 2014*); (b) its accompanying press releases; (c) a worldwide interest by the international media, with articles in more than 15 languages (some media enumerated here: https://peerj.com/articles/297/#links); (d) the wide dissemination of the paper, which was open-access and available from the publisher's website, PubMed Central, Academia.edu, and ResearchGate; (e) a "question" asked on the social networking site ResearchGate (https://www.researchgate.net/post/Have_you_seen_this_land_planarian_an_invasive_alien_species); (f) for France, a citizen science survey, based on a blog (*Justine, 2014*) and a twitter account (https://twitter.com/Plathelminthe4); (g) for New Caledonia (including the Loyalty Islands), a general survey officially organized by PB, which involved 140 technical partners from local administrations, research institutions and plant nurseries (see Table 1).

Photographs of *P. manokwari* (Figs. 1–4) were obtained from various sources (Table 1).

### Fixation of worms

For fixation, two protocols were used: either (a) the living specimen was simply put into room-temperature ethanol (80–95%); or (b) the living worm was put into boiling water; after 1–2 min water was removed and replaced with room-temperature ethanol (80–95%). No narcotizing agents were used. Specimens of *P. manokwari* (Table 1), collected by hand

Justine et al. (2015), *PeerJ*, DOI 10.7717/peerj.1037

**Table 1** Origin of *Platydemus manokwari* specimens and observations.

| Locality | Collector and/or photographer | Photos | Date (dd/mm/yyyy) | Specimen deposition (H: histology) | COI Sequence (GenBank #) | COI Haplotype |
|---|---|---|---|---|---|---|
| **France** | | | | | | |
| Caen, hothouse in Jardin des Plantes | FREDON Basse Normandie | | 29/10/2013 | MNHN JL81 | Short: KF887958 (identical in 3 specimens) Long: KR349594 | World |
| Caen, hothouse in Jardin des Plantes | FREDON Basse Normandie | | 15/04/2014 | MNHN JL139 (H) | Long: KR349597 | World |
| **French Polynesia** | | | | | | |
| Fa'a'ā, Tahiti, altitude: sea level | Jonas Fernandez | + (Fig. 4) | 26/05/2014 | MNHN JL151 | Short: KR349595 | World |
| Moorea, Tahiti, Mt Aorai trail, altitude 1,000 m | Jean-Yves Meyer | + | 19/06/2013 | | | |
| **Wallis and Futuna** | | | | | | |
| Wallis, Uvéa, near Lac Lalolalo | Jean-Yves Meyer | + | 10/11/2007 | | | |
| Futuna, locality not registered | Jean-Yves Meyer | + | 10/11/2008 | | | |
| **New Caledonia, mainland** | | | | | | |
| Nouméa, quartier Vallée des Colons | Nicolas Rinck | + | 09/04/2014 | MNHN JL107B | Short: KR349596 | World |
| Nouméa, quartier N'Géa | Claire Goiran | + (Fig. 3) | 29/07/2014 | | | |
| Nouméa, quartier N'Géa | Claire Goiran, specimens CEN#2500-2501 | | 25/12/2014 | MNHN JL221A MNHN JL221B | | |
| Nouméa, quartier Motor Pool | Vanessa Héquet | | 06/09/2014 | | | |
| Mont-Dore, Saint Michel | Ludivine Sariman, specimen CEN#2496 | | 01/10/2014 | MNHN JL197 | Short: KR349600 | World |
| Païta, Tontouta | Gazmira Machin-Baucher, specimen CEN#2497 | | 12/10/2014 | MNHN JL198 | Short: KR349601 | World |
| Païta, Ondémia | Patrick Barrière, specimens CEN#2502-2503 | + | 08/01/2015 | MNHN JL222A MNHN JL222B | | |
| Hienghène, quartier Pai Kalone | Cyrille Sabran, specimens CEN#2504-2505 | | 26/01/2015 | MNHN JL223A | | |
| La Foa, quartier Nily | Jörn Theuerkauf, specimen CEN#2506 | | 21/02/2015 | MNHN JL234 | Long: KT004666 | World |
| Koné, Foué (CEN) | Patrick Barrière, specimens CEN#2507 | | 27/02/2015 | MNHN JL235A | Long: KT004667 | World |
| Koné, Foué (CEN) | Patrick Barrière, specimens CEN#2594 | | 03/04/2015 | | | |
| Koné, Paiamboue | Nathalie Baillon, specimen CEN#2596 | | 11/04/2015 | | | |
| Koné, Tribu de Tiaoué | Hervé Vandrot, specimens CEN#2508 | | 02/03/2015 | MNHN JL236A MNHN JL236B | Long: KT004668 Long: KT004669 | World World |
| Koné, Village, rue Paul Amat | Ken Cadin, specimen CEN#2510 | | 05/03/2015 | MNHN JL238 | Long: KT004671 | World |
| Dumbéa, La Couvelée | Béatrice Bresse, specimen CEN#2509 | | 02/03/2015 | MNHN JL237 | Long: KT004670 | World |
| Poya, Lot 16, Section village | Nicolas Bazire, specimen CEN#2592 | + | 08/03/2015 | | | |
| Poya, Népou | Lory Richard, specimen CEN#2591 | | 12/03/2015 | | | |

Table 1 (*continued*)

| Locality | Collector and/or photographer | Photos | Date (dd/mm/yyyy) | Specimen deposition (H: histology) | COI Sequence (GenBank #) | COI Haplotype |
|---|---|---|---|---|---|---|
| **New Caledonia, Lifou Island** | | | | | | |
| Tribu de Hnassé | Jean-Paul Lolo, specimen CEN#2593 | | 03/03/2015 | | | |
| **New Caledonia, Maré Island** | | | | | | |
| Tribu de Maré, Limite | Marcel Pijone, specimen CEN#2595 | | 11/04/2015 | | | |
| **Singapore** | | | | | | |
| Opera Estate, Fidelio Street | Andrew Wee Kien Han | + (Fig. 2) | 20/03/2014 | MNHN JL148A | Long: KR349579 | World |
| | | | | MNHN JL148B | Long: KR349580 | World |
| Chestnut Avenue | Personal blog[*], photos by "James" | + | 05/02/2011 | | | |
| | Personal blog[*], photo by "Ivan Kwan" | + | 17/07/2010 | | | |
| Tanah Merah | Personal blog[*], photo by "Ivan Kwan" | + | 18/04/2011 | | | |
| Admiralty Park | Personal blog[*], photo by "James" | + | 22/07/2010 | | | |
| Secondary forest, end of Sunset Way | Benjamin Paul Yi-Hann Lee | + | 01/01/2014 | | | |
| **Solomon Islands** | | | | | | |
| Guadalcanal, Foxwood, east of Honiara | Crispus Fanai | + | 08/2014 | | | |
| Guadalcanal, Henderson, east of Honiara | Crispus Fanai & Francis Tsatsia, specimen examined by LW, LW1804) | | 08/2014 | MNHN JL199 | Short: KR349602 | Australian 1 |
| | | | | MNHNJL200A | Short: KR349603 | Australian 1 |
| | | | | MNHN JL200B | Short: KR349604 | Australian 1 |
| | | | | MNHN JL200C | Short: KR349605 | Australian 1 |
| | | | | MNHN JL200D | Short: KR349606 | Australian 1 |
| Guadalcanal, Dodo Creek, east of Honiara | Crispus Fanai & Francis Tsatsia, specimen examined by LW, LW1805 | | 09/2014 | MNHN JL201A (H) | Long: KR349593 | Australian 1 |
| | | | | MNHN JL201B (H) | Short: KR349607 | Australian 1 |
| | | | | MNHN JL201C | Short: KR349586 | World |
| | | | | MNHN JL201D | Short: KR349608 | World |
| | | | | MNHN JL201E | Short: KR349609 | Australian 1 |
| | | | | MNHN JL201F (H) | Long: KR349588 | World |
| | | | | MNHN JL201G (H) | Long: KR349589 | World |
| | | | | MNHN JL201H | Long: KR349590 | World |
| | | | | MNHN JL201I | Long: KR349591 | World |
| | | | | MNHN JL201J | Long: KR349592 | World |
| Guadalcanal, West Honiara, Tasahe Drive | Bob Macfarlane | + | 05/04/2015 | | | |
| **Guam** | | | | | | |
| Quenga turnoff on Sengsong Road (28) | David G. Robinson, specimen USDA #: 04-GU-11 | | 17/08/2004 | MNHN JL191 | Not obtained | |
| Alongside road to Ritidian Point | David G. Robinson, specimen USDA #: 05-GUAM-15 | | 22/08/2005 | MNHN JL192 | Not obtained | |

Justine et al. (2015), *PeerJ*, DOI 10.7717/peerj.1037

Table 1 (*continued*)

| Locality | Collector and/or photographer | Photos | Date (dd/mm/yyyy) | Specimen deposition (H: histology) | COI Sequence (GenBank #) | COI Haplotype |
|---|---|---|---|---|---|---|
| **USA, Florida** | | | | | | |
| Miami, NW 5th Avenue | Mary Yong Cong, specimen DPI#: WP#2 | | 10/08/2012 | MNHN JL189 | Short: KR349598 | World |
| Miami, SW 122 Street | Anibal Altamirano, specimen DPI#: none | | 06/09/2012 | MNHN JL190 | Short: KR349599 | World |
| Miami, SW 192 Terrace | Mary Yong Cong & Juan Suarez, DPI#: none | | 30/10/2014 | USDA 140203 | | |
| Coral Gables, Montgomery Botanical Gardens, Old Cutler Road | Makiri Sei | + (Fig. 1) | 14/08/2014 | | | |
| **USA, Puerto Rico** | | | | | | |
| San Juan | Giomara La Quay | + | Dec/2014 | MNHN JL207A | Short: KR349610 | World |
| | | | | MNHN JL207B | Short: KR349611 | World |
| **Australia, Queensland** | | | | | | |
| Townsville, Palmetum, Douglas | Leigh Winsor, specimens LW1795 | | 17-09-2014 | MNHN JL179A | Long: KR349583 | Australian 1 |
| Townsville, Palmetum, Douglas | Leigh Winsor, specimens LW1795 | | 17-09-2014 | MNHN JL179B | Long: KR349584 | Australian 1 |
| Townsville, Condon | Leigh Winsor, specimen LW1796 | | 17-09-2014 | MNHN JL180 | Long: KR349585 | Australian 1 |
| Townsville, Condon | Leigh Winsor, specimens LW1794 | | 28-10-2009 | MNHN JL178B | Long: KR349582 | Australian 1 |
| Townsville, Condon | Leigh Winsor, specimens LW1794 | | 28-10-2009 | MNHN JL178A | Long: KR349581 | Australian 2 |

**Notes.**

* Available from: http://lazy-lizard-tales.blogspot.co.uk/2011/12/ribbons-terrestrial-nemerteans-of.html (maintained by Ivan Kwan).

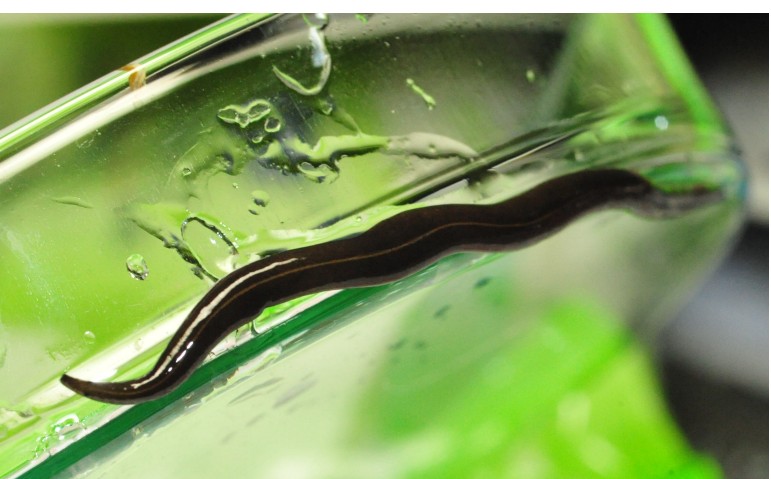

**Figure 2** *Platydemus manokwari* **in Singapore.** Photograph by Andrew Wee Kien Han.

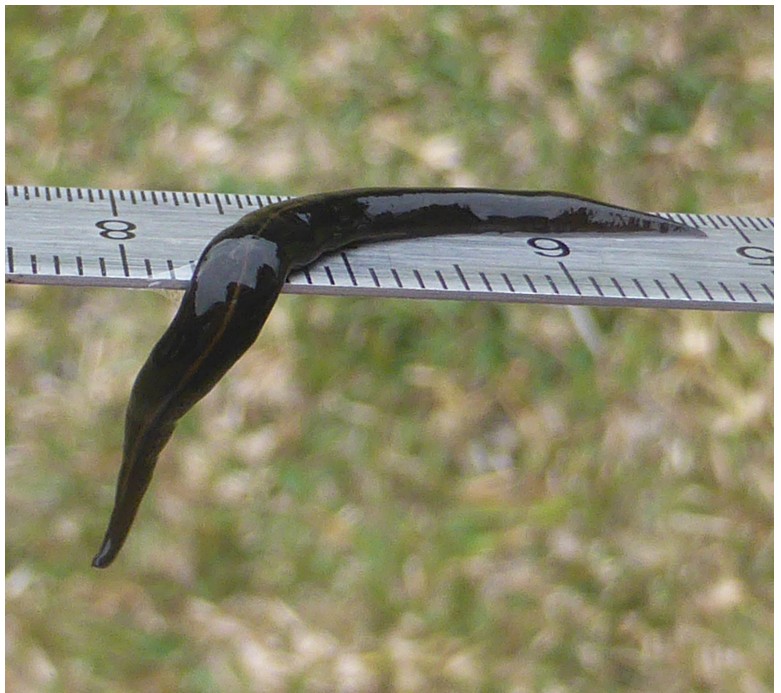

**Figure 3** *Platydemus manokwari* **in Nouméa, New Caledonia.** Photograph by Claire Goiran. Scale: cm and mm.

and fixed, were sent to Paris by postal service. In addition, we collected new specimens from the infested hothouse in Jardin des Plantes, Caen, France and in public and private gardens in Townsville, Australia. Specimens were deposited in the collections of the Muséum National d'Histoire Naturelle, Paris, France (MNHN), and the United States Department of Agriculture, Animal and Plant Health Inspection Service (USDA APHIS) National Malacology Laboratory, Academy of Natural Sciences, Philadelphia, USA.

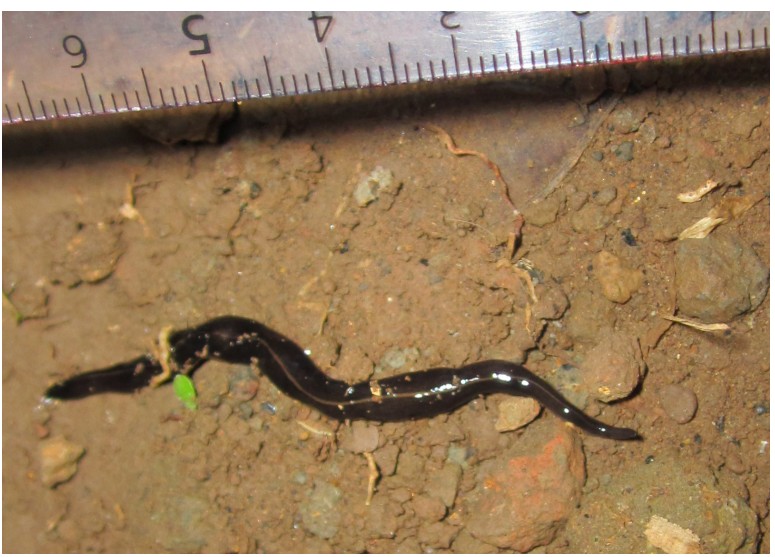

**Figure 4** *Platydemus manokwari* **in Fa'a'ā, Tahiti, French Polynesia.** Photograph by Jonas Fernandez. Scale: cm and mm.

**Table 2** **Morphological data of specimens with known haplotype.** Specimens with "World haplotype" apparently differ from specimens with "Australian haplotype" only by the ratio length of penis papilla: length of body. Given our small sample size we conclude that no morphological difference can reliably differentiate the two haplotypes.

| Specimen | Haplotype | Length of body (mm) | Length of penis papilla (µm) | Ratio length of penis papilla: length of body (%) |
|---|---|---|---|---|
| MNHN JL139 | World | 44.5 | 639 | 1.4 |
| MNHN JL201F | World | 22.5 | 355 | 1.6 |
| MNHN JL201G | World | 18+ (1–2 mm of tip missing) | 320 | 1.7 |
| LW 1065 (used for GenBank AF178320) | Australian | 45.0 | 315 | 0.7 |
| MNHN JL201A | Australian | 28.0 | 213 | 0.7 |
| MNHN JL201B | Australian | 26.5 | 256 | 0.9 |

## Anatomical analysis

In order to determine whether the two main haplotypes could be recognized morphologically the dimensions of various characters were measured in three sexually mature specimens of the World haplotype (MNHN JL139, 201F and 201G all from Dodo Creek, Solomon Islands), and in three specimens of the Australian haplotype (GenBank AF178320/LW1065 from Townsville, and MNHN JL201A and 201B from Dodo Creek, Solomon Islands). The posterior portion of mature specimens that were selected for both molecular analysis and histology (Table 2) were given a second change of absolute ethanol, then transferred to xylene, and finally infiltrated and embedded in Paraplast® Tissue Embedding Medium. Tissue blocks were sectioned at 8 µm in the longitudinal sagittal plane using a rotary microtome, then affixed to slides with glycerine-egg albumen, stained using the trichrome Picro-Gomori method (*Menzies, 1959*), and mounted in D.P.X with colourfast (Fronine, Riverstone, Australia).

## Molecular sequences

For molecular analysis, a small piece of the body (1–3 mm$^3$) was taken from the lateral edge of ethanol-fixed individuals. Genomic DNA was extracted using the QIAamp DNA Mini Kit (Qiagen). Two sets of primers were used to amplify the COI gene. A fragment of 424 bp (designated in this text as "short sequence") was amplified with the primers COI-ASmit1 (forward 5′-TTTTTTGGGCATCCTGAGGTTTAT-3′) and COI-ASmit2 (reverse 5′-TAAAGAAAGAACATAATGAAAATG-3′) (*Littlewood, Rohde & Clough, 1997*). The PCR reaction was performed in 20 μl, containing 1 ng of DNA, 1× CoralLoad PCR buffer, 3Mm MgCl2, 66 μM of each dNTP, 0.15 μM of each primer, and 0.5 units of Taq DNA polymerase (Qiagen). The amplification protocol was: 4′ at 94 °C, followed by 40 cycles of 94 °C for 30″, 48 °C for 40″, 72 °C for 50″, with a final extension at 72 °C for 7′. A fragment of 825 bp was amplified with the primers BarS (forward 5′-GTTATGCCTGTAATGATTG-3′) (*Álvarez-Presas et al., 2011*) and COIR (reverse 5′-CCWGTYARMCCHCCWAYAGTAAA-3′) (*Lázaro et al., 2009*), following (*Mateos et al., 2013*). PCR products were purified and sequenced in both directions on a 3730xl DNA Analyzer 96-capillary sequencer (Applied Biosystems). Results of both analyses were concatenated to obtain a COI sequence of 909 bp in length (designated in this text as "long sequence"). Sequences were edited using CodonCode Aligner software (CodonCode Corporation, Dedham, MA, USA), compared to the GenBank database content using BLAST and deposited in GenBank under accession number KR349579–KR349611, KT004666, KT004667, KT004668, KT004669, KT004670 and KT004671. A newly obtained sequence of *Parakontikia ventrolineata*, GenBank KR349587, from a specimen collected in Brest (France) and kept in the MNHN collection as MNHN JL95, was used as outgroup for the NJ tree. For several specimens only "short" sequences were obtained (Table 1). No sequence was obtained for the specimens from Guam (Table 1), which were kept in ethanol for a decade.

## Trees and distances

MEGA6 (*Tamura et al., 2013*) was used to estimate genetic distances (*p*-distance and also kimura-2 parameter distance) and the evolutionary history (Figs. 5 and 6) was inferred from the kimura-2 parameter distance using the Neighbour-Joining method (*Saitou & Nei, 1987*); all codon positions were used.

# RESULTS

## Analysis of molecular data

We obtained COI sequences from 38 individuals. Of these, 21 were "long" sequences, obtained with two pairs of primers, and 17 were "short" sequences, when only one pair of primers provided results.

The analysis of the 38 "short" sequences showed that only two haplotypes were present. One haplotype was found in specimens from France, French Polynesia, New Caledonia, Singapore, Florida and Puerto Rico, and certain specimens from the Solomon Islands; we designate it as the "World haplotype". One haplotype was found only in specimens

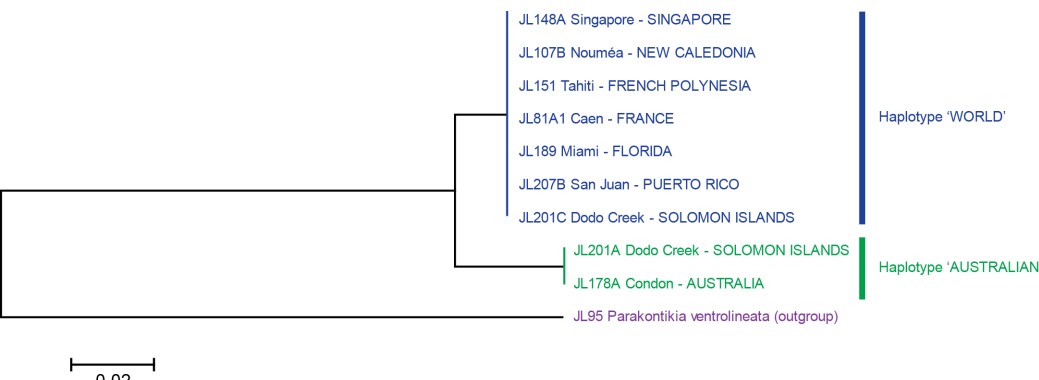

**Figure 5** *Platydemus manokwari*: **Tree based on short COI sequences.** 'Short' sequences, 424 bp in length, were obtained from 38 specimens. In this tree, only one sequence was used for each locality or for each haplotype when variation was found (Dodo Creek, Solomon Island). Two clades are well differentiated: haplotype "World" and haplotype "Australian". The tree was constructed using the Neighbour-Joining method. The scale bar indicates the number of substitutions per site.

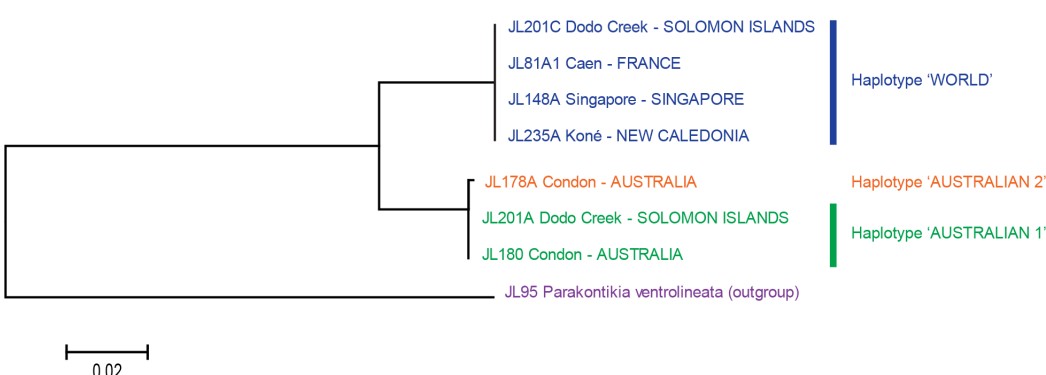

**Figure 6** *Platydemus manokwari*: **tree based on long COI sequences.** 'Long' sequences, 909 bp in length, were obtained with two pairs of primers from 21 specimens. In this tree, only one sequence was used for each locality or for each haplotype when variation was found. Two clades are well differentiated: haplotype "Australian" (with a minor 1 bp variation in one sequence, labelled as "Australian 2") and haplotype "World". The tree was constructed using the Neighbour-Joining method. The scale bar indicates the number of substitutions per site.

from Australia and certain specimens from the Solomon Islands; we designate it as the "Australian haplotype". The difference between the two haplotypes was 15 bases, on a total of 401 (*p*-distance: 3.7%; kimura-2 parameter distance: 3.8%); no individual variation of these "short" sequences was found in any of the two haplotypes. A tree constructed from these sequences shows two well differentiated branches, with identical sequences in each of the branches (Fig. 5).

The analysis of the 21 long sequences also revealed the same two haplotypes, World and Australian. The difference between the two haplotypes was 42 bases, on a total of 848 (*p*-distance: 4.8%; kimura-2 parameter distance: 5.0%). A minor variation was found in the Australian haplotype, with one specimen having a one-base difference (distance from this specimen to the World haplotype: *p*-distance: 5.0%; kimura-2 parameter distance: 5.2%); this haplotype is designated as "Australian 2" and the other, major haplotype is

designated as "Australian 1". No variation was found in the World haplotype (15 sequences from four countries) for these long sequences. A tree constructed from these sequences shows two well differentiated branches, with identical sequences in the World haplotype branch and minor variation in the Australian haplotype branch (Fig. 6).

Variation within localities was strikingly different between Dodo Creek, Solomon Islands and the rest of the studied localities. All specimens from France (3), New Caledonia (9 from different localities), Singapore, Florida and Puerto Rico (2 in each case) were identical and displayed the World haplotype; all specimens from Townsville had the Australian haplotype with a minor difference of one base for one specimen (haplotype Australian 2). Sequences from Henderson (Solomon Islands) were identical in the five specimens, with haplotype Australian 1. In contrast, the ten specimens from Dodo Creek included three animals with haplotype Australian 1 and seven with haplotype World. The Solomon Islands were thus the only country in which genetic variation among specimens was observed.

## Morphological identification

The flatworms (Figs. 1–4) presented the following morphological characteristics: body broadest in the middle, tapering evenly anteriorly but more abruptly posteriorly; two large prominent eyes back from the tip of the elongate snout-like head; dorsum a dark olive brown colour; pale cream median dorsal longitudinal stripe beginning just behind the eyes and continuing to the posterior tip; olive brown colour grading to grey at anterior tip; two thin submarginal cream stripes with fine lower greyish margin running laterally from the anterior end along the length of the body; ventral surface a pale finely mottled light brown. These features are consistent with those of *Platydemus manokwari* de Beauchamp, 1963 (Platyhelminthes, Continenticola, Geoplanidae, Rhynchodeminae) (*de Beauchamp, 1962*; *Justine et al., 2014*; *Kawakatsu, Ogren & Muniappan, 1992*; *Winsor, 1990*).

The dimensions and characters that were compared between specimens of the two haplotypes included body length and width, the distance of the mouth and gonopore from the anterior end, the lengths of the oesophagus, pharynx and pharyngeal pouch, distance between the common sperm duct and common ovovitelline duct, distance from the common sperm duct to the tip of the penis papilla, penis length and penis width at it base, length of the male atrium, length of the female atrium, length of the glandular canal, length of the common ovovitelline duct and depth of the viscid gland. Sexually mature specimens of *P. manokwari* from the Solomon Islands were found to be considerably smaller (22–28 mm long) than those from Caen, France (44.5 mm long) and Townsville, Australia (45 mm long). The only characters that were found that may possibly differentiate between the two haplotypes are the length of the penis papilla expressed as a percentage of the total body length (Table 2), and the shape of the penis papilla. Specimens of the World haplotype generally exhibited a finger-shaped or elongate conical penis papilla that was 1.4–1.7% of the body length, similar in appearance of the copulatory organs of *P. manokwari* figured by de Beauchamp (Figs. 1A and 1B in *de Beauchamp, 1962*) whereas in specimens of the Australian haplotype the penis papillae tended to be a low regular conical shape with a length less than 1% of the total body length (0.7–0.9% of

the body length), closely resembling the copulatory organs of *P. manokwari* figured by de Beauchamp (Fig. 1 in *de Beauchamp, 1972*) and by *Winsor (1990)* (Fig. 1 in *Winsor, 1998*).

## Time of invasion and other remarks

In France, after the first finding of *P. manokwari* (October 2013), the hothouse of the Jardin des Plantes in Caen in which the species was found was closed and public access restricted. In April 2014, a thorough search was performed in the same hothouse. Several species of land planarians were found (some are still unidentified), including four adult specimens of *P. manokwari*. A citizen-based survey of land planarians in France (*Thévenot, Justine & Rome, 2014*), supported by important media involvement (list of links: https://peerj.com/articles/297/#links), yielded hundreds of records of non-indigenous land planarians in France (*Justine, 2014*; *Justine, Thévenot & Winsor, 2014*) but never a record of *P. manokwari* in the wild.

The presence of the species in Singapore had already been noted in 2010, according to unpublished observations listed in Table 1, and the species is probably spreading; according to the observations of one of us (AWKH), the species was not present in his garden before 2013. Local observations show that *P. manokwari* predates the introduced giant African snail, *Achatina fulica* Bowdich, 1822.

The detection of *P. manokwari* in New Caledonia is recent (2014). After the discovery of the first specimens in the capital city, Nouméa, in April 2014, a survey was conducted from September 2014 to April 2015. Specimens were found in several places in the mainland, including Southern and Northern Provinces, and in two of the three Loyalty Islands, Maré and Lifou. The current extensive distribution of such a cryptic species, evidenced after only eight months of survey, suggests that the introduction of *P. manokwari* is not recent. Moreover, discussions with scientists involved in the observation of soil species revealed that the flatworm had been observed several years ago, but no substantiated proof of these observations could be found. Since a number of endemic land planarians exist in New Caledonia (*Schröder, 1924*; *Winsor, 1991*), it could not be ascertained if these earlier observations were actually of *P. manokwari*, or of other species. *P. manokwari* was not detected when large surveys were conducted in the 1990s (*Gargominy et al., 1996*)—it is thus likely that its introduction occurred in the last 20 years. Local observations show that *P. manokwari* predates introduced *A. fulica*.

In Wallis and Futuna, our records of *P. manokwari* are based only on two photographs taken by one of us (JYM), in 2007 in Uvea (Wallis Islands) and in 2008 in Futuna (Hoorn Islands). These two island groups are separated by more than 100 km of sea. No more recent records are available.

The presence of *P. manokwari* on Guam has been known since 1977 (*Eldredge & Smith, 1995*; *Hopper & Smith, 1992*). Our records, dated 2004, confirm the continuing presence of the species; to our knowledge, no more recent records are available.

In the USA, the accidental introduction of *P. manokwari* through human agency to Florida is probably recent, with our first specimens found in August 2012. The species is apparently now well established, with several different locations found in 2014 in Miami

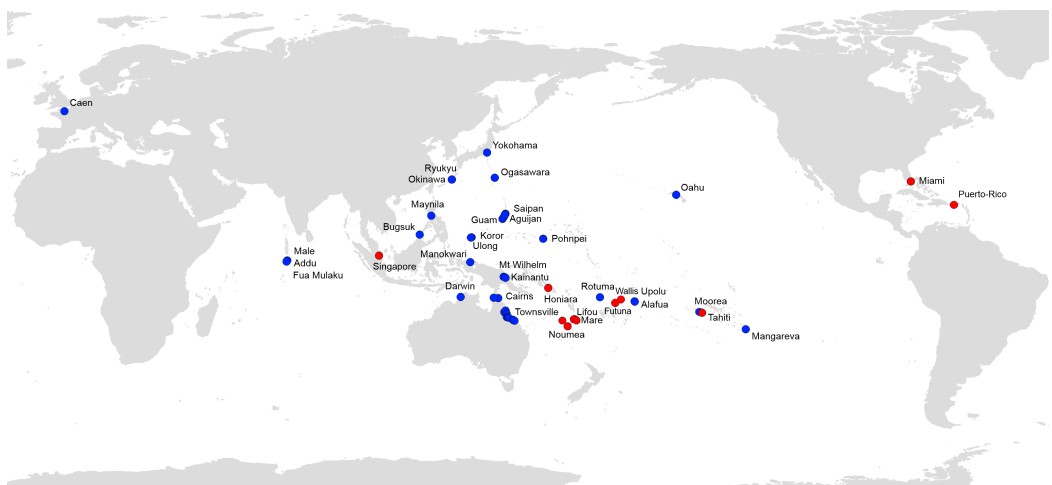

**Figure 7** *Platydemus manokwari,* **map of distribution records.** Blue: previous records (*Justine et al., 2014*); Red: new records reported in this paper.

Dade County (Table 1). The introduction of *P. manokwari* to Puerto Rico is also probably recent, with the first specimens found in December 2014, in a single locality, San Juan (Table 1). To our knowledge, no survey aiming specifically at detecting *P. manokwari* has been done in other parts of Florida or in other US states. Land planarians are often photographed by amateurs; however, a survey of various websites and various Google searches (December 2014) did not reveal any photographic report of *P. manokwari* in the US.

The introduction of the species to Guadalcanal, Solomon Islands also seems recent (2014). Specimens of *P. manokwari* were found in fields invaded by the introduced Giant African snail, *Achatina fulica*. Although the report of these flatworms in Guadalcanal is recent, we do not know if this is truly an indication of a recent introduction or not. The case of New Caledonia, where the species was found in many localities after a coordinated participative survey was initiated, and, on the contrary, the case of Guam, for which we have no report for the last decade, show that land planarians are detected and reported only when an appropriate survey is undertaken. It is obvious that land planarians in Solomon Islands, a country with low income and expensive internet access, are less often photographed by amateur photographers who post their findings on the internet than they are in the US or Singapore.

Figure 7 shows all available records of *P. manokwari* in the world. The spread of the species mainly concerns countries and territories in the Pacific Area; Florida and Puerto Rico are the first records in the Americas. The record in Florida is the first on the American mainland.

## DISCUSSION

### Morphological and molecular identification of specimens

The external morphological characteristics of the specimens found in the new locations, Singapore, Tahiti, New Caledonia, Wallis and Futuna, Solomon Islands, Florida and Puerto Rico, are similar to *Platydemus manokwari* from other locations. Histological examination

of sexual specimens confirmed the identity of the species in Caen, France, the Solomon Islands, and Townsville, Australia.

The length and shape of the penis papillae may possibly reflect the relative maturity of the specimens (young sexual specimens compared to relatively older sexual specimens); unfortunately there are presently no data available for *P. manokwari* correlating the age of sexual specimens with the continuing development of the copulatory organs. Alternatively the length and shape of the penis papillae may in fact reflect differences between these two haplotypes. Based on observations of the partly extended and contracted penis of various specimens of the land planarians *Parakontikia ventrolineata* and *Kontikia mexicana*, it was concluded that "unless conditions of fixation are absolutely standard, penial protrusion must be regarded at best as an unreliable character" (*Jones, Johns & Winsor, 1998*). All the specimens that were investigated in this study were killed without the use of a narcotizing agent and were fixed in ethanol. In the specimens examined histologically, the penis papilla was either the elongate finger-like conical shape or the low regular conical shape; there appeared to be no intermediary forms. Recently, the form of the penis papilla was included in the suites of characters used to morphologically distinguish between cryptic species revealed by molecular analysis (*Álvarez-Presas et al., 2015*), where in the copulatory organs of *Obama decidualis* the penis papilla was irregularly shaped, sometimes conical, otherwise much folded, compared to *O. anthropophila* in which the penis papilla had a regular conical form. However given the very small sample size in our study we hesitate to unequivocally state that the length and form of the penis papilla reliably differentiate one haplotype of *Platydemus manokwari* from the other.

Sequences of COI comprised two haplotypes, "Australian" (with a minor variation on a single nucleotide in a single specimen among 13) and "World" (France, New Caledonia, Singapore, Tahiti, Florida, Puerto Rico; all identical in 19 specimens). The two haplotypes were found together in the same locality only in the Solomon Islands. The difference between the two haplotypes was 3.7% (computed on short sequences) and 4.8% (computed on long sequences). In the literature there are reports of variations in the intraspecific and interspecific genetic distances of species in the same family as *P. manokwari*, i.e., the Geoplanidae, For *Microplana* spp., specimens considered to belong to the same species varied by up to 4%, and specimens from different species varied by 19% (*Álvarez-Presas et al., 2012*). In other genera, some species show high intraspecific values, up to 8.3% (*Lago-Barcia et al., 2015*). The difference of 4.8% between the two variants of *P. manokwari* found here could be either considered as relatively high intraspecific variation, or as evidence for the presence of two different species. In view of the limited morphological and anatomical differences found between specimens with known haplotypes, and the small size of our sample, we provisionally conclude that a single species, *Platydemus manokwari*, is involved.

## The world distribution of the two COI haplotypes

The Solomon Islands contrast with all other localities in the world for which we have molecular data for *P. manokwari*: in all other localities a single haplotype was detected.

Moreover, the same haplotype (which we named "World") was found in localities as distant as Solomon Islands, French Polynesia, New Caledonia, Singapore, Tahiti, France, Florida and Puerto Rico. This suggests that the species encountered a bottleneck of genetic diversity in the early stages of its human-mediated dispersal throughout the Pacific region and that all specimens found in these distant countries and territories come from an original population with low genetic variation.

How do we interpret the presence of two haplotypes in the Solomon Islands? *Platydemus manokwari* was described from Papua New Guinea (*de Beauchamp, 1962*) and was found in several localities in this large island, from the coast to an altitude of 3,000 m (*de Beauchamp, 1962*; *de Beauchamp, 1972*). We have, unfortunately, no information about the genetic structure of these populations, and our knowledge of the genetic variation in other places is based on a single marker, the COI gene. The Solomon Islands are less than 1,000 km away from the Eastern part of Papua New Guinea. Human contacts between these islands might be as ancient as 30,000 years (*Sheppard, 2011*). For these reasons, it is either possible that *P. manokwari* was inadvertently transported from Papua New Guinea by early Melanesian settlers or that the invasion is extremely recent. The presence of both haplotypes in Solomon Island could be an indication that several populations were transported there from Papua New Guinea or Australia, at dates unknown. It remains that other countries and territories were invaded with two populations of *P. manokwari*, one with no genetic variation, which has been transported to many localities (the "World" COI haplotype) and one, with very low variation, which is found only in Australia (the "Australian" COI haplotype).

## Significance of these new records—a threat to biodiversity

As far as we are aware, these records are the first of *P. manokwari* in Singapore, New Caledonia, Wallis and Futuna, the Solomon Islands, the Caribbean, and North America. They thus add six countries and territories to the list of 16 territories recently published, which included Irian Jaya, New Guinea, Australia, Guam and Northern Mariana Islands, Philippines, Japan, Maldives, Palau, Hawaii, Federated States of Micronesia, French Polynesia, Samoa, Tonga, Vanuatu, Fiji and France (*Justine et al., 2014*).

In French Polynesia, records were known only from Moorea Island (*Lovenburg, 2009*) and Mangareva Island (*Justine et al., 2014*; *Purea et al., 1998*); the present records are the first from Tahiti Island, the main island of French Polynesia. However, Moorea is very close to Tahiti (15 km) with significant inter-island traffic via several ferries, and it is not surprising that the two islands share the species. Of interest is that the species was recorded in Tahiti Island in two stations with different altitudes, sea level and about 1,000 m. This is the only locality for which we have altitude records; most other records are from close to sea level, though the species has been recorded from sea level to 3,000 in Papua New Guinea (*de Beauchamp, 1962*; *de Beauchamp, 1972*).

In New Caledonia, it was demonstrated by one of us (PB) that when a coordinated survey is organized, the knowledge about the distribution of a relatively cryptic species can be dramatically increased in a matter of months; although the first record in a private

garden was made in April 2014, we have now confirmed records of *P. manokwari* in more than 15 locations, including both Southern and Northern Provinces and two of the Loyalty Islands. It is therefore clear that the species has been inadvertently spread to most of the provinces and main islands of New Caledonia.

In France, the current situation is that *P. manokwari* is confined to a single hothouse in the Jardin des Plantes in Caen, but is not eradicated. Eradication is still an issue of concern, because the species could be a major threat to various soil invertebrates, especially snails, including endemic species (*Justine et al., 2014*).

*Platydemus manokwari* has now been found in 22 countries and territories in the world, mainly in the Pacific area (Fig. 7). It is thus clear that the genetic results presented here (from 8 countries and territories) represent only a fraction of these disjunct populations of the worm. A survey of the genetics of populations in the area of origin (Papua New Guinea) is necessary to fully interpret the results, as are the use of additional genetic markers; these are beyond the scope of this paper which relies on a single marker (COI) on a limited number of samples, with none from the country of origin. Our discovery of the existence of two haplotypes is potentially important for our understanding of the invasion by *Platydemus manokwari*, with, apparently, one haplotype more successful than the other.

Furthermore, while most of the infected countries and territories reported until now are islands, from which the spread of the species through human agency is limited by means of transportation and various business and biosecurity protocols, our new record, Florida, will not be subjected to these limitations. In addition to their natural spread, specimens of *P. manokwari* can easily be passively spread mainly with infested plants, plant parts and soil. The species could potentially eventually be spread from Florida throughout the US mainland, and this can be considered a significant potential threat to the whole US, the West Indies and even the rest of the Americas.

## ACKNOWLEDGEMENTS

Many individuals (listed in Table 1) are thanked for forwarding photographs of the flatworms for identification. Gary Barker provided technical advices for the survey in New Caledonia. Michelle Tanguy provided the specimens of *P. ventrolineata*. Nicolas Puillandre (MNHN) helped with the analysis of the molecular results. The reviewers are thanked for their valuable help. We also thank hundreds of French and European citizens who sent photographs of non-indigenous land planarians, among which none were *P. manokwari*.

### Funding

MNHN ATMs "Barcode" and "Emergences" provided financial support for molecular analysis. The funders had no role in study design, data collection and analysis, decision to publish, or preparation of the manuscript.

## Grant Disclosures

The following grant information was disclosed by the authors:
MNHN ATMs "Barcode" and "Emergences".

## Competing Interests

Jean-Lou Justine is an Academic Editor for PeerJ.

## Author Contributions

- Jean-Lou Justine conceived and designed the experiments, performed the experiments, analyzed the data, contributed reagents/materials/analysis tools, wrote the paper, prepared figures and/or tables, reviewed drafts of the paper, initiated and ran citizen science program about land planarians in France.
- Leigh Winsor conceived and designed the experiments, performed the experiments, analyzed the data, contributed reagents/materials/analysis tools, wrote the paper, prepared figures and/or tables, reviewed drafts of the paper, performed the histological analysis.
- Patrick Barrière contributed reagents/materials/analysis tools, prepared figures and/or tables, reviewed drafts of the paper, organized a survey in New Caledonia, examined collected specimens and photographs.
- Crispus Fanai, Andrew Wee Kien Han, Giomara La Quay-Velázquez and Francis Tsatsia contributed reagents/materials/analysis tools, prepared figures and/or tables, reviewed drafts of the paper, provided specimens and photographs.
- Delphine Gey performed the experiments, contributed reagents/materials/analysis tools, reviewed drafts of the paper, performed molecular analyses.
- Benjamin Paul Yi-Hann Lee contributed reagents/materials/analysis tools, prepared figures and/or tables, reviewed drafts of the paper, provided photographs.
- Jean-Marc Lefevre and David Philippart contributed reagents/materials/analysis tools, prepared figures and/or tables, reviewed drafts of the paper, organized a survey in the hothouse in Caen, provided specimens and photographs.
- Jean-Yves Meyer contributed reagents/materials/analysis tools, reviewed drafts of the paper.
- David G. Robinson contributed reagents/materials/analysis tools, prepared figures and/or tables, reviewed drafts of the paper, requested specimens and gathered information from Florida.
- Jessica Thévenot contributed reagents/materials/analysis tools, prepared figures and/or tables, reviewed drafts of the paper, made maps.

## DNA Deposition

The following information was supplied regarding the deposition of DNA sequences:
GenBank: KR349579–KR349611, KT004666, KT004667, KT004668, KT004669, KT004670 and KT004671.

## Data Deposition

The following information was supplied regarding the deposition of related data:

All figures will be deposited in FigShare and Wikimedia.

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
