# Peer review of "The invasive land planarian Platydemus manokwari (Platyhelminthes, Geoplanidae): records from six new localities, including the first in the USA"

_PeerJ, doi:10.7717/peerj.1037_

## Round 0.1 · original submission · Major Revisions

· Academic Editor

Major Revisions

Overall I think that this paper will make an important contribution to the literature. However, I do think that you need to consider it to be a stand-alone piece of work (rather than just a brief follow-on from the previous paper) and therefore you need to expand greatly on several aspects (see the reviewers comments below and my comments, which I have to email to you separately). In addition, there are several pertinent comments from the reviewers about the analysis of the molecular data that you need to address (e.g., Bayesian trees vs NJ). In addition, the writing needs some tidying up so that the paper is more comprehensive and understandable. In particular, you should develop a clear argument in your introduction, including specific research questions and hypotheses that are then addressed in the results and discussion.

Reviewer 1 ·

Basic reporting

No comments.

Experimental design

In "Molecular sequences":
" The PCR reaction was performed in 20 μl, containing 1 ng of DNA, 1× CoralLoad PCR buffer, 3Mm MgCl2, 66 µM of each dNTP, 0.15μM of each primer, and 0.5 units of Taq DNA polymerase (Qiagen)" (line 75)... is it correct 1 ng of DNA? Usually we use the minimum of 20-50 ng of DNA...

Validity of the findings

About NJ tree... I suggest a Bayesian analysis because NJ tree, although widely used, is not as robust as the Bayesian .

Additional comments

You need to review the "References". "Saitou & Nei, 1987" and Tamura et al., 2004" were not mentioned in the text.
The pictures reminded me Geoplana francisca Leal-Zanchet & Carbayo 2001.

·

Basic reporting

No comments.

Experimental design

No comments.

Validity of the findings

No comments.

Additional comments

This paper is relevant in many aspects. First, by the enlargement of the geographical distribution of P. manokwari, the only flatworm considered among the 100 most dangerous invasive species of the world, which is not a minor detail. Second, although it is expected (by its invasive ability) its occurrence near its possible area of origin, its record for the first time in America is a concern. So, if appropriate measures are not taken, is expected soon to record this species in other countries in the Americas and other continents. The authors provide for the first time, even partially, the genetic structure of this invasive species.
I consider the manuscript is well written, is clear and concise, and provides novel and useful information. Therefore, I recommend its publication. The authors will find some minor comments and corrections along the manuscript.
I agree that it is very important to quickly communicate the presence of species considered invasive, because it can accelerate control measures. However, it is essential to have as much evidence as possible to identify the species, including internal morphological data. It would be also interesting to compare the internal morphology of the "two variants" that have been found.

Reviewer 3 ·

Basic reporting

The ms has an extremely short introduction and no objectives described. In fact the paper seems to be only intended to present new records of an introduced terrestrial planarian species in areas where it has not been found before. A part from that information, the only scientific data included are some short and long sequences from the captured animals, data used to give support to the assignment of the individuals to P. manokwari (when compared to the sequences present in Genbank, some obtained in a previous work by the same authors). Finally the authors make some inferences on the possible invasion times of the different areas basing exclusively on the records collected from bibliography or from the own authors.
In the discussion the authors state that other information on the species has been published previously in this same journal and that other interesting studies as finding out the relationship of the invader flatworms to individuals coming from its original area of distribution (Papua New Guinea) are out of the scope of the paper.
In consequence, from my point of view the ms offers little new information, I’m not sure this falls within the scope of the journal (giving a list of new, and interesting, records of an invasive species). However I understand from the policy of the journal that this may be acceptable provided the correctness and robustness of the methodologies and conclusions reached, so I give in the following sections some comments on the methodologies used which I think need to be corrected, and the results/conclusions reached.

Experimental design

The material and methods section dedicated to the molecular analyses is very poor. It repeats twice that MEGA6 has been used to do the analyses (line 88 and line 93), but does not explain in much detail what has been done, which can be found in different places. For example, part of the information lacking in the material and methods is given in the figure legend for the phylogenetic tree, which does not seem to me an adequate place. In the figure legend also appear the total length of the tree and the number of positions used which will be better in the results section.
As for the analyses, the authors have used Kimura-2 parameters to calculate genetic distances per se, but used the Maxiumum Composite Likelihood method to calculate the distances to construct a NJ tree. They do not state why using one or the other or why not using the same distance correction for both analyses.
The authors also state that 1st+2nd+3rd+Noncoding were included to construct the tree, having sequenced a fragment of COI it is clear that no noncoding regions can be included, for clarity to any reader in the ms it must be stated simply that all codon positions have been used for the analyses.
The legend of the tree figure also gives some extra information in a pair of sentences that are terribly complicated:
“The tree is drawn to scale, with branch lengths in the same units as those of the evolutionary distances used to infer the phylogenetic tree. The evolutionary distances were computed using the Maximum Composite Likelihood method (Tamura et al. 2004) and are in the units of the number of base substitutions per site.”
To explain that it is enough to state that scale bar is given in base substitutions per nucleotide.
On the other hand, given the few differences between the two clades of P. manokwary, the null differences within them, and that the authors have not included any other species closely related nor individuals from a variety of localities in the original area of distribution, it seems not necessary to include a tree in this study since it does not give any extra information to the obtained with the genetic distances per se. However, I understand it is an easy way to show the results, so in case the authors want to keep thg tree please have into account my previous comments.
Another question is the denomination of the sequences obtained for the two groups as “Australian variant” and “Other variant”, it will be a lot more scientific to call them haplotypes, simply numbering them or giving them a name.
To infer the NJ tree the authors use the short fragment sequenced, this is 424 bp long but the alignment included 317 sites, which is more than 100 less than the short fragment length, but it is not explained the reason to have discarded so many sites. Maybe this is due to the fact that the two fragments overlap only partially and long sequences do not completely include the short sequence? This have to be clarified, or it should be explained the reason and criteria to remove so many sites for the analysis.

Validity of the findings

The identification of the species seems robust enough. The big difference in distances found with the two fragments (3.8% short and 5.2% long, it seems that the short fragment includes a more conserved region of the COI gene) puts a certain stress over stablishing comparisons with other studies, unless it is completely sure that the same fragment has been studied. Anyway, in this case the differences do not seem to affect the conclusion of the two clades belonging most probably to the same species (but, as the authors recognize, more studies must be done to reach a definitive conclusion on this issue).

However, in the same way as there was no an “original question” (no objectives), I think there are neither specific conclusions for the ms a part from the statements that the species have a broader “invasive” distribution than known before.
The rest of the discussion includes a hypothesis that is only one possible explanation, among many, of the observed data. The hypothesis puts forward that the data may be explained by two invasions from two different populations of the original area of distribution, but it is also possible that both invasions come from a very diverse population in the original area of distribution, that the animals arriving to Australia included some variability while the “others” passed through a greater bottleneck, or maybe the new variant appeared in Australia..… There are other possibilities that could be checked through population genetic analyses and phylogenetic studies including animals from the original region, etc. But the authors say this is out of the scope of the paper, hence I think this paragraph must be removed if the ms is accepted.
The section “significance of these new records” is quite a max mix, it explains all the places where the species has been recorded for the first time (again), there is a sentence stating it is very interesting that the species has been found at two very different altitudes in one island, a data not mentioned before and not given for any other locality in table 1. It would be interesting that the authors explain why it is so interesting to mention that.
Lines 213-215 seem out of place, why eradication is still an issue of concern? Has something been done yet to eradicate?
Lines 216-217, are repetitive or should better be at the beginning of the section when the description of all records is done. And from line 217 to 222, seems better to go together with the final remarks in line 231 to the end
Lines 223 to 229 are conclusions already reached in previous papers, and then the sentence in line 230 will be better suited in this paragraph and not in the following.

---

## Round 0.2 · Minor Revisions

· Academic Editor

Minor Revisions

The reviewers and I all feel that only minor revisions are necessary (mostly rewording), but please pay particular attention to the detailed comments from Reviewer 3, who gives you a helpful guide for obtaining the correct results. I will send the manuscript back to this reviewer once you have resubmitted it, so please respond in full to all recommendations. Please also thank your reviewers in the acknowledgements section.

Reviewer 1 ·

Basic reporting

This new version has a more complete introduction.
No Comments.

Experimental design

I accepted your explanation about the analysis.
No Comments.

Validity of the findings

No Comments.

Additional comments

I consider this new version is better than the first. The manuscript is well written, is clear and concise, and provides novel and useful information. It is very important to quickly communicate the presence of species considered invasive in order to accelerate control measures. I recommend its publication.

·

Basic reporting

No comments.

Experimental design

In "Material and methods", the section "anatomycal analysis" seems to be a bit short. Indeed, the second paragraph of results (in "morphological identification", lines 148-157) belong to the section M & M.

Validity of the findings

No comments.

Additional comments

The section "results" starts with the analysis of molecular data. Perhaps, it could start with the morphological identification, to follow a logic sequence with "material and methods" and "discussion".

In results, it would be interesting to add a figure with histological sections of the copulatory apparatus of two specimens of different haplotype. This would be helpful in order to compare with futures findings of this species in other regions, since this is highly expected.

In discussion, the last paragraph of the section "Morphological identification of specimens" (lines 235-247) would fit better in a separate section, like "analysis of molecular data", since is not dedicated to morphological aspects.

The last part of the discussion (Significance of these new records – a threat to biodiversity) sounds a bit long and repetitive with section "Time of invasion and other remarks". Perhaps it could be shortened.

Minor corrections along the manuscript are included in the pdf version.

Reviewer 3 ·

Basic reporting

This is a second revision, the general impressions is that the ms is better than the first version but there are still some problems with the methodologies used. The authors have incurred in some mistakes that I feel is compulsory that they solve before the ms is published in a journal that is going to be read by many people and that can be used by some as example of how a certain analysis has to be done.

Experimental design

It is compulsory that the authors do changes on the phylogenetic inference section.
It is true that a NJ tree is more than enough for what they want to show and also that MEGA6 is an adequate program to run such an analysis. However one has to be careful because user-friendly programs may allow doing things that are not correct, and the authors in this case have done some mistakes in applying the methodology.
In the first place to run a phylogeny on distances, as is the case of NJ, one has to calculate a distance and better if also applies a correction (as a Kimura-2 parameters, the more general used). But they have used the number of differences method (an absolute number of differences), as they explain in the material and methods section and can be seen in the tree figures where the scale shows a 5 and a 20 respectively. This is wrong. Although the general aspect of their tree is not going to change much, it is important that any analysis published in a good journal is accurate since anyone can copy the methodology and in others cases this can make a big difference, and as I stated it is wrong, neighbour-joining as all distance methods are prepared to work on genetic (or whatever) distances not absolute numbers of changes.
So, it is necessary that the authors redo this analysis (which will take them 10 minutes: in the phylogeny inference select NJ and then select for example Kimura-2p distance to infer the tree; redoing the tree as a nice picture will take some more time, but it will be no more than a pair of hours).
Also it is better that they deselect the box indicating “non coding” sites in that window. Since they are working only with a coding region they are only dealing with 1st, 2nd and 3rd codon sites, so there are no “non coding” sites. However, again a user-friendly program is prepared to use those types of sites, and so it offers the possibility. If the authors do not deselect it then it appears on the figure legends the program prepares automatically and it does not make any sense.
As for the sentences included in the figure legend given automatically by MEGA6, that now the authors have used as the material and methods, there is a terrible and difficult to follow sentence stating: “The trees were drawn to scale, with branch lengths in the same units as those of the evolutionary distances used to infer the phylogenetic tree. The evolutionary distances were computed using the number of differences method (… ) and are in the units of the number of base differences per sequence”
This sentence is constructed by the program as the sentences of GPS navigators, by putting together sentences already prepared and adding some information: the first sentence is exactly the same in all cases only stating that the branch lengths are in the units of the evolutionary distance used, and then comes the sentence that the program adds in each case, stating which evolutionary distance has been used in each case (which in the case of this ms was not an evolutionary distance but the number of substitutions), and then comes another automatic sentence “and are in the units of” and then again the specific case. This is terribly difficult to read. If the authors now use the kimura-2p as I propose in the material and methods they simply need to state:
“The program MEGA6 was used to estimate genetic distances (p-distance and also Kimura-2 p) and the evolutionary history was inferred from the kimura-2p distances using the N-J method (ref) all codon positions were used.”

To give the information on the tree branch lengths they only have to state in the tree figure legend that “the scale bar indicates the number of substitutions per site” (if they use a genetic distance the value over the scale bar will be below 1).

In the material and methods they also state that “all ambiguous positions were removed from each sequence pair”. Again here MEGA gives a not totally clear information. It offers the possibility of removing from all the alignment (complete deletion) or for each sequence pair (pairwise deletion) the gaps and missing data. If you select the first the caption automatically written will say: “all positions containing gaps and missing data were eliminated”; But if you select the second it says “all ambiguous positions were removed from each sequence pair”. The term ambiguous here is certainly equivocal, it is better that the authors write “all positions containing gaps and missing data were removed from each sequence pair” or simply "the pairwise deletion option was selected". Ambiguous can be used to refer to polymorphic sites but also to those regions in an alignment for which the alignment is not sure, so in the case of MEGA it is better to specify that they refer to gaps and missing data (N’s) only.

Validity of the findings

The findings are valid

---

## Round 0.3 · accepted · Accept

· Academic Editor

Accept

Thank you for making all the changes suggested by the reviewers. I am happy that the analyses are now correct and it is good to see that it doesn't change your result.